# Cell Adhesion Molecules in Normal Skin and Melanoma

**DOI:** 10.3390/biom11081213

**Published:** 2021-08-15

**Authors:** Cian D’Arcy, Christina Kiel

**Affiliations:** Systems Biology Ireland & UCD Charles Institute of Dermatology, School of Medicine, University College Dublin, D04 V1W8 Dublin, Ireland; cian.darcy@ucdconnect.ie

**Keywords:** cadherins, GTEx consortium, Human Protein Atlas, integrins, melanocytes, single cell RNA sequencing, selectins, tumour microenvironment

## Abstract

Cell adhesion molecules (CAMs) of the cadherin, integrin, immunoglobulin, and selectin protein families are indispensable for the formation and maintenance of multicellular tissues, especially epithelia. In the epidermis, they are involved in cell–cell contacts and in cellular interactions with the extracellular matrix (ECM), thereby contributing to the structural integrity and barrier formation of the skin. Bulk and single cell RNA sequencing data show that >170 CAMs are expressed in the healthy human skin, with high expression levels in melanocytes, keratinocytes, endothelial, and smooth muscle cells. Alterations in expression levels of CAMs are involved in melanoma propagation, interaction with the microenvironment, and metastasis. Recent mechanistic analyses together with protein and gene expression data provide a better picture of the role of CAMs in the context of skin physiology and melanoma. Here, we review progress in the field and discuss molecular mechanisms in light of gene expression profiles, including recent single cell RNA expression information. We highlight key adhesion molecules in melanoma, which can guide the identification of pathways and strategies for novel anti-melanoma therapies.

## 1. Introduction

Cell adhesion molecules (CAMs) are critical for the formation and structural integrity of many tissues, including the epithelium of human skin [1]. Broadly speaking, there are two types of cell adhesion molecules, those that mediate cell–cell attachment and those that facilitate cellular interactions with the extracellular matrix (ECM) [2]. However, there are four main protein families that facilitate the cell–cell and cell-ECM binding (Table 1). First is the integrin family, which mediates cell-ECM connections as their main function, though some members also facilitate cell–cell binding. Second, members of the cadherin family, which are calcium-dependent and mediate adherens junction formation. Third, the group of immunoglobulin-like cell adhesion molecules (IgCAMs). The fourth group is the selectin family. Under physiological conditions, CAMs facilitate primarily the adhesion of the cell to other cells or to the ECM. Moreover, they have are involved in a myriad of cellular functions, such as cell signalling, cell maintenance, and cell development [2].

The melanocyte, a neural crest derived cell, resides on the basement membrane (BM) of the basal epidermis at a ratio of 1:10 with the surrounding keratinocyte cells. They are responsible for the protection of around 35 of the surrounding keratinocytes from ultraviolet (UV) radiation via the secretion of melanin, which absorbs harmful UV rays [3]. The key to maintaining homeostasis in this tightly regulated layer of the skin is the presence of cell adhesion molecules.

Melanoma is an aggressive form of skin cancer that is often fatal in the later stages of the disease. Melanoma is characterised by the transformation of melanocytes into melanoma cells [3]. There are three main histological types of cutaneous melanoma in patients: superficial spreading, nodular and lentigo maligna. Despite recent advances in the treatment of the disease, such as the combination of immunotherapies and checkpoint inhibitors, there remain high relapse rates and a large non-responsive population of melanoma patients [13]. Given the heterogeneity between patients, there may be multiple reasons a therapy does not cause a response or shows a different efficacy between patients. The microenvironment in which the tumour resides, the tumour microenvironment (TME), can influence the efficacy of the immunotherapies and the intracellular targeted therapies such as BRAF inhibitors [13]. The TME is an altered niche where the melanoma cells reside. It is characterised by multiple features, such as a hypoxic environment (reduced oxygen supply), an increased presence or lack of immune cells, an altered metabolism, an increase in reactive oxygen species (ROS), a changed physical environment as well as ECM deposition and modified acidity [14].

CAMs are important in the context of melanoma as they can influence the ability of the immune system to gain proximity to, and recognise the tumour. They also allow the tumour to migrate and metastasise. Indeed, from the early stages of melanoma, there is a change in the CAM expression profile, which begins with a loss of E (epithelial)-cadherin (cadherin 1; *CDH1*) expression and associated loss of communication with the regulatory keratinocytes in the epidermis [15,16]. A shift to N (neuronal)-cadherin expression (cadherin 2; *CDH2*) allows the melanoma cells to preferentially bind to fibroblast cells that also express this CAM, and thus promotes invasion into the dermis below. The CAM expression profile changes again when the cells become metastatic and start to express, in many cases, CD146 (MUC18; *MCAM*) allowing the cell–cell adhesion to distant cells [17]. The expression of these CAMs can be altered due to the TME and vice-versa [2]. This CAM alteration can determine survival of the cells in a new microenvironment in early melanomagenesis, the ability to invade different tissues, and the efficacy of treatment. Understanding the dynamics of the change in CAM expression in the TME will be important in the treatment of melanoma, and the analysis of mRNA and protein expression data of CAMs in physiologic skin and melanoma will help to elucidate new avenues of research.

Large scale and high coverage ‘OMICS’ datasets of protein and gene expression levels in human healthy and cancerous tissues are ever increasing (e.g., GTEx [18] and TCGA (https://www.cancer.gov/tcga) (accessed on 4 June 2021)), and data are provided in comprehensive databases, such as the Human Protein Atlas (https://www.proteinatlas.org/ (accessed on 4 June 2021)) [19]. Furthermore, it is exciting that expression levels in individual cell types within the tissue are also becoming increasingly available—thanks to advances in single-cell RNA sequencing (scRNA-seq). For example, recently scRNA-seq data of human skin [20] and metastatic melanoma [21] were accomplished. These expression data, in particular when stratified for specific (melanoma) patient groups, together with molecular mechanistic studies, are the enabling factors for a successful implementation of a personalised medicine framework [22].

This review focuses on CAMs that are important for physiological skin homeostasis, but also emphasises those CAMs associated with the melanocyte to melanoma transition and melanoma progression. We provide a comprehensive list of the CAMs known to be implicated in melanoma, CAMs where the microenvironment has been shown to have a role in the changing CAM expression profile and, thus, the fate of the melanocyte/melanoma cell. Further, we highlight potential alternative targets for increasing current therapeutic efficacy and/or elucidating potential research avenues for novel therapeutic targets in melanoma.

## 2. The Cadherin Family Mediating Cell–Cell Attachment

Cadherins, the calcium-dependent adhesion protein family, are molecules that facilitate the adherence of cells via adherens junctions or desmosomes. Cadherins are some of the most common cell–cell adhering molecules, with 115 cadherin family members present in the human genome, which can further be divided into the classical (four members), desmosomal (seven members), protocadherin (72 members), and unconventional (32 members) sub-families (Appendix A). Cadherins usually operate in a homotypic fashion, though they can also adhere heterotrophically [23] (Figure 1A). The extracellular binding region of members of the cadherin family consists of 5 extracellular calcium (EC1–EC5) dependent repeat regions that allow for strong binding, mainly facilitated by EC1 and EC2 [24]. They gain their adherence strength via an intracellular structural region, which binds to actin filaments in the cell. The cadherin family plays two major roles in the physiological skin, by connecting cells together in the epidermis (e.g., melanocytes and keratinocytes), and by facilitating the binding of cells to the BM, in a desmosome cell-BM interaction (along with plakoglobin and desmoplakin) [25]. The desmosome interactions are known as the spot welds of cell–cell adhesion and are important in the epithelial and cardiac structures allowing for strength in mechanically involved tissue [26].

Expression changes of the cadherin family of CAMs are well-recognised during the development of melanocytes in the neural crest, where the melanoblasts, following an increase in WNT signalling (which induces differentiation), detach their cadherin anchors from the neural crest and migrate along the neural crest pathway. They eventually reach the epidermis where they are reattached to the surrounding ECM and cells via the re-expression of cell adhesion molecules such as E-cadherin [4]. N-Cadherin and E-cadherin expression is reduced in melanoblasts before their migration along the dorsolateral pathway to the epidermis. In mice, the endothelin receptor B and its ligand endothelin B are required for melanoblast migration, along with the KIT receptor and its ligand KITL [4]. The decrease of cytoskeletal organisation proteins, such as RAC1, also seems to be important in the migration phase, however mice knockouts (KO) of these proteins have mild effects on development, suggesting that these are not completely essential for the migration of the melanoblasts from the neural crest. It was also shown in mice that, once expressed in the epidermis, within 48 h the levels of E-cadherin rose 200-fold, establishing the connection with the melanocytes and the epidermis [4]. The interplay of microenvironmental factors and cell adhesion in this instance leads to a physiological event, important in the development of the epidermis. This interplay can, however, lead to disease when it is perturbed. Indeed, the loss of cell adhesion molecule expression due to genetic abnormalities or microenvironmental changes has been long associated with disease [27].

### 2.1. Cadherins in Physiological Skin Context

E-cadherin is present under physiological conditions in the skin connecting the melanocyte to the neighbouring keratinocytes at a fixed ratio (one melanocyte per 10 keratinocytes; “epidermal melanin unit”), which is important for epidermal homeostasis as the keratinocytes regulate the growth of the melanocytes. In the physiological epidermal melanin unit, the expression of E-Cadherin on the melanocytes inhibits the RhoA pathways and thus the proliferation and growth of the melanocyte via the intracellular p120 protein (*CTNND1*) bound to E-cadherin. p120 is a catenin protein that recruits microtubules to the cadherin binding complex leading to a mature junction [28]. E-cadherin has also been shown to play a role in cell signalling and can stabilise β-catenin (*CTNNB1*) by binding the transcription factor intracellularly and releasing it when required. N-cadherin mediated cell–cell adhesion is essential for the development of the central nervous system [29]. In the context of the skin, N-cadherin expression is found, under physiological conditions, on fibroblasts in the dermis allowing for their cell-to-cell binding. Recent studies have shown that fascia fibroblasts conduct their cell migration for wound repair via upregulation of N-cadherin, leading to swarming of fibroblast cells and a pathological fibrotic response or scar formation. The inhibition of N-cadherin leads to reduced scaring via reduced fascia mobilization [30].

### 2.2. Cadherins in Pathological Skin

The loss of E-cadherin is an early event in most melanomas. This event usually occurs either as a (epi)-genetic mutation or because of protein disruption or as a change in the TME, which may induce epithelial to mesenchymal transition (EMT) [16]. The loss of E-cadherin is associated with the downstream activation of TWIST, Snail, Slug, and SIP1 transcription factors that bind the E-boxes in the E-cadherin promoter and supress its expression, while also being associated with proliferation, metastasis and chemoresistance [16,31]. The ability of the melanoma cells to undergo EMT and develop a more invasive phenotype can also occur through the upregulation and expression of matrix metalloproteinase 2 (MMP2). MMPs are proteases that normally function in the epidermis to facilitate tissue remodelling and repair, whereas in cancer they facilitate the passage of cells through the ECM [32]. This can degrade E-cadherin on the melanocyte/melanoma cell and the surrounding ECM leading to increased mobility [5]. As mentioned above in physiological skin, the p120 bound to the intracellular E-cadherin inhibits proliferation. The loss of E-cadherin also leads to the loss of p120’s tumour suppressor functions, allowing cellular proliferation of the early melanoma cells to increase via the release of β-catenin or the activation of Rac1 and the MAPK pathway [28]. Recently it was shown that in the absence of UV exposure, the decrease of desmoglein 1 (*DSG1*) in keratinocytes increased the production of the melanocyte stimulating hormone pro form (pro-opiomelanocortin, *POMC)* and, thus, induced an increase in melanin production in the melanocytes. The loss of desmoglein 1 in keratinocytes and the loss of control of melanocytes may serve as a niche that contributes to the early loss of melanocyte control in melanomagenesis [33]. In melanoma genesis, the desmosome structures are vital for the adherence of melanocytes to the BM and these are lost following the loss of E-cadherin expression, facilitating the cells’ ability to migrate [26]. A recent study suggests that melanomas that either express or contain subpopulations of E- cadherin expressing melanomas have favourable outcomes with combination immune checkpoint blocking (ICB) therapy [34]. The expression of E-cadherin in metastatic melanoma is likely due to the decreased signalling of EMT pathways, leading to a partial or full expression of E-cadherin. The authors initially investigated the possibility of an increase in β-catenin as the reason for loss of resistance. However, when whole lysate quantitative mass spectrometry was performed on E-cadherin negative and exogenously amplified E-cadherin in a highly ICB resistant melanoma cell line, there was no difference in the levels of β-catenin. Upon further investigation, they found an increase in CD103 (αE β7 integrin) [34]. Many CAMs can play important roles in the progression of melanoma though they may not be expressed on any skin cells, they may be expressed on immune cells and blood vessel epithelial cells, which can also influence the progression of melanoma. Thus, the expression of CAMs in the TME is as important as their expression in melanoma cells. Due to the re-established expression of E-cadherin on the melanoma cells, the CD103 + T-cells were able to attach to the melanoma cells. This was corroborated by the presence of E-cadherin and CD103 positive T-cells as demonstrated by immunohistochemistry in samples from patients that responded to ICB treatment [34]. Therefore, it seems E-cadherin is not only important in the initial phases of melanoma, keeping it from leaving the epidermis, but it also seems to play a role in attracting T-cells and allowing them to carry out their cytotoxic antitumour function. This also highlights a cooperative role of the cadherin and integrin families in homeostatic immune response in the epidermis.

N-cadherin is normally expressed on mesenchymal cells and is upregulated in many melanomas (Figure 1B and Figure 2A). Following the loss of E-cadherin, N-cadherin expression increases in melanomagenesis [35]. This increase allows the preferential binding of the melanoma cells to the dermal fibroblasts over the keratinocytes as N-cadherin can trans/cis bind to the growth factor receptor on fibroblasts and activate pathways associated with invasiveness, and thus there is further loss of control of melanocyte/melanoma growth [6]. This event is known as “cadherin switching” and E/N-cadherin are not the only cadherin members to switch, but it is the most relevant switch in the context of melanoma. The TME has been associated with EMT in melanoma and the cadherin switching. Stromal cells such as fibroblasts can secrete growth factors such as transforming growth factor beta (TGF-β), WNT, and NOTCH ligands, and hepatocyte growth factor (HGF) as the microenvironment contributes to the EMT and E/N-cadherin switching. These secreted factors promote the SNAIL–TWIST transcription factor activation, which as mentioned previously is associated with the loss of E-cadherin and initiation of EMT. These transcription factors can also be activated by miRNA and degraded by E3 ubiquitin ligases, thus can be alternatively activated by the secreted factors from the stromal cells [36].

N-cadherin has also been recently explored as a therapeutic target for melanoma. A study by Ciołczyk-Wierzbicka and Laidler [7] found that the inhibition of N-cadherin led to the decrease of MMP2, which decreased the migratory phenotype of the cells. The authors also suggested that the overexpression of N-cadherin was activating the PI3/AKT mTOR and MAPK/ERK kinase pathways leading to increased proliferation [7]. Because of this role of N-cadherin in melanoma progression, it was tested as a target in a clinical trial in 2011. Using ADH-1, a selective and competitive N-cadherin inhibitor that disrupts the adhesion ability of N-cadherin, the study showed that despite a sensitizing of the tumours to chemotherapy there was no significant benefit in the response to the treatment over time [37].

### 2.3. Gene Expression Profiles of Cadherins in Normal Skin and Melanoma

Among the 115 cadherin family members, 90 genes are expressed in healthy human skin (Appendix A) [19,20]. E-cadherin is among the most highly expressed genes within melanocytes and keratinocytes in normal skin (Appendix A), while N-cadherin is barely expressed at all (0.2 pTPM, protein Transcripts per million mRNA measurement unit). Other highly expressed genes are desmoglein-1, desmocollin-3 (*DSC3*), desmocollin-1 (*DSC1*), cadherin-related family member 1 (*CDHR1*), calsyntenin-1 (*CLSTN1*), desmoglein-3 (*DSG3*), and protocadherin Fat 2 (*FAT2*). Gene expression data in individual cell types show that the top 20 most highly expressed genes are generally highest expressed in melanocytes, with exception of cadherin-13 (*CDH13*) (highest expression in endothelial cells), protocadherin-1 (*PCDH1*) (highest expression in smooth muscle cells), protocadherin gamma-C3 (*PCDHGC3*) (highest expression in fibroblasts), and cadherin-11 (*CDH11*) (highest expression in fibroblasts) (Figure 2B). It is interesting to note that the cadherins that have reduced protein expression levels in melanoma (Figure 2A) tend to be mainly expressed on melanocytes and keratinocytes and are approximately balanced in skin (Figure 2B) (e.g., desmoglein-1, desmocollin-3, desmocollin-1, calsyntenin-1, desmoglein-3), reinforcing that a loss of cell adhesion between melanocytes and keratinocytes is a critical event in melanoma. In contrast, cadherin genes that are highly expressed in melanoma (Figure 1B and Figure 2A) tend to have low expression in normal skin in keratinocytes (e.g., calsyntenin-1 and cadherin-3), but their expression in endothelial cells, macrophages, and fibroblasts (e.g., protocadherin-1) may enable ‘wrong’ or ‘unphysiological’ cell–cell contacts promoting invasion and metastasis.

In the cadherin family many of the members are lost or downregulated in melanoma. PCDHGC3, however, is barely expressed in normal melanocytes (Appendix A) but is highly expressed in melanoma (Figure 1B and Figure 2A), therefore the cadherin family members are not exclusively downregulated or lost in melanoma.

## 3. Integrins in Cell-ECM Adhesion

The integrin family consists of 27 integrin isoforms (Appendix A). Integrins are cell adhesion molecules that assist in the anchorage of the cell by mediating cell-ECM adhesion, cell–cell adhesion, and migration of cells. Integrins by contrast to the cadherin family are calcium-independent cell adhesion molecules. Integrins find their specificity for binding through their extracellular domain (Figure 3A). In the binding of a cell to the ECM the binding to fibronectin can be facilitated by α5β1 (ITGB1), αVβ3 (ITGB3), etc., collagen by α1β1 (ITGA1), etc., Laminin α2β1 (ITGA2). They can also interact with the immunoglobulin superfamily CAMs, ICAM (ICAM1) interacts with αLβ2 (ICAM3), while αMβ2 (ITGAM) and VCAM (VCAM1) with α4β1 (ITGA4) [38].

### 3.1. Integrins in Physiological Skin Context

Movement by mesenchymal cells via lamellipodia is important during the development process, such as in the migration of melanoblasts/melanocyte cells from the neural crest to the epidermis [4]. A similar mechanism occurs during wound healing [39]. Cells move following integrin activation in a polarised direction into the ECM, which acts as a physical support for migration. The movement of cells depends on the differing regulation of various integrins on multiple parts of the cell at the same time. From the leading edge of the cell to the rear, there is a difference in the affinity of the integrin binding, the valency, and the dynamics of the integrin interaction. At the leading edge of the cell, there is high affinity, and once an integrin ligand is bound, there is a rapid clustering of integrin interactions and increase in valency in the front of the cell [40]. This is essential in stabilizing the actin scaffold in the filopodia and the lamellipodia, and RAC1 mediates this process through actin polymerization [41]. The integrin binding at the rear of the cell must be dynamic with its interactions with the ECM and is moved forward towards the high affinity binding at the front of the cell, which is largely mediated by RhoA and actin reorganization [38]. In the meantime, at the tail end of the cell, the integrins reduce their binding affinity and there is a dissolution of the adhesion complexes and, hence, detachment from the ECM, allowing the cell to move forward in the polar direction on the ECM protein vitronectin. Adaptor proteins and enzymes, such as FAK, Src, and integrin-linked kinase (ILK) lead to further downstream activation that modify the cytoskeleton, and promote migration, as a result many of these adaptor proteins are upregulated in cancer [42].

The connection of the integrin proteins on the melanocyte cell and most epithelial cells to the BM allows for the response to growth and survival factors. This connection is also responsible for the cytoskeletal organization, shape, and polarity of the cell. The integrin connection usually prevents the cell from undergoing apoptosis. Thus, if a cell migrates from the correct integrin interactions or loses its integrin binding, it undergoes anoikis, which is a cell death mechanism. Physiologically, anoikis is important for the shedding of cells in the epidermis, allowing for cell renewal, but it also prevents cells from detaching from their physiological microenvironment and interacting with a distal microenvironment [43]. Therefore, the integrin interaction on the cell with the ECM essentially instructs the behaviour of the cell, and this loss or alteration is of great importance in the progression of melanoma.

### 3.2. Integrins in Pathological Skin

Similar to normal conditions, where mesenchymal cells or cells involved in wound repair migrate, melanoma cells migrate and metastasise around the body using the ability of the integrin ECM binding to facilitate movement [38]. Melanoma cells cannot sufficiently migrate after losing E-cadherin expression alone, they acquire an altered integrin expression profile based either on genetic changes or as a response to a new environment [44,45] (Figure 3B and Figure 4A). In melanoma, integrins are associated with a switch from the initial radial growth pattern of the disease to the vertical growth pattern, associated with a poorer overall patient prognosis [46].

Integrins are also implicated in angiogenesis, where they mediate signalling through the secretion of growth factors in the microenvironment that promote vessel growth [47]. The acquisition of the integrin αVβ3 (*ITGB3*) for example, is seen exclusively on metastatic melanoma but not on benign naevi. Depending on crosstalk with different receptor tyrosine kinases, integrin αVβ3 can affect different cell fates, such as proliferation, survival, and metastasis [8,9]. Integrin αVβ3 also plays a role in angiogenesis via the reciprocal activation of the vascular endothelial growth factor receptor 2 (VEGFR2) as well as the survival and maturation of newly formed blood vessels. This integrin/angiogenesis link in melanomas is important as the αVβ3 inhibition in melanoma cells leads to the NRP-1, a coreceptor of VEGF-A being inhibited, thus reducing the αVβ5/NRP-1 dependent angiogenesis in melanoma [46].

Increased integrin production, however, is still insufficient for metastasis because of the dense network or fibrils that compose the ECM. The acquisition of proteinases, such as metalloproteinases, serine proteases, and cathepsins is required for the remodelling of the ECM to allow the cell to move through its network. MMPs promote the functions of integrins in the TME [48]. The breakdown of the ECM, primarily by MMP-2 in melanoma causes the exposure of integrin-ECM binding sites in the ECM that can increase migration, but also the cleavage of other cell surface proteins (e.g., cadherins) that can increase the cellular signal to metastasise [44,49]. This step is crucial in the melanoma cells breakdown of the BM and entrance into the dermis.

Integrins primarily facilitate the connection of the cell to the surrounding ECM. Under physiological conditions, the anchorage provided by integrins to the ECM is so important that upon loss of this interaction anoikis is triggered. Through the upregulation of MCL-1, melanocytes/melanoma cells can survive detaching from integrins in the basal epidermis [50]. Integrin αVβ3 expression in melanoma allows for the remodelling of the surrounding ECM. Melanoma cells shape their microenvironment by secreting proteases, which have been shown to denature collagen bound to integrin αVβ3 and increase the BCL-2:BAX protein ratio five-fold. This allows the melanoma cells to survive in a new environment instead of undergoing anoikis [51]. The ability to enhance survival through integrin αVβ3 expression is through the downstream activation of FAK and PAK1 via AKT, ERK activation and the activation of β-catenin and mobility though LIMK [41]. Other CAMs, such as MUC1 (*MUC1*), a cell–cell adhesion molecule implicated in lung metastasis in melanoma, have also been shown to resist anoikis in other epithelial cancers [52].

Integrins, such as integrin αVβ3, which are expressed in melanoma cells that typically metastasise to the lungs [46], have recently been suggested as drivers of a melanoma stem like phenotype in K-RAS melanoma tumours. Integrin αVβ3 is a promising target that is overexpressed only on melanoma cell surfaces, and hence offers a degree of selectivity in therapy. However, a phase 2 clinical trial targeting integrin αVβ3 with Cilengitide failed to provide benefit [53]. In a recent study the authors found that together with KRAS and galectin 3, melanoma cells could induce a stem-like phenotype and resistance to chemotherapeutics via a TBK1 activated NFκ-B signalling pathway, which is involved in the activation of inflammation [54,55]. As the disruption of NFκ-B has previously led to toxicity when treating lung cancer, the authors sought to inhibit integrin αVβ3 simultaneously with a specific inhibitor and cilengitide (a prototypic integrin inhibitor). Combination treatment using a αVβ3 specific inhibitor and chemotherapeutics inhibited the melanoma cell growth and reversed the stem cell like phenotype [55]. These recent studies have shown that targeting integrins in melanoma has the potential to deliver therapeutic benefit, and given the involvement of integrins in melanoma and their under-representation in the literature they are attractive targets for future research.

### 3.3. Gene Expression Profiles of Integrins in Normal Skin and Melanoma

The integrin family consists of 27 members, of which transcripts for all genes are detected in normal human skin—albeit some with low levels (Appendix A) [19,20]. Among the most highly expressed genes in normal skin, 10 integrin members are very highly expressed (*ITGB4*, *ITGB5*, *ITGB1*, *ITGA5*, *ITGA3*, *ITGAV*, *ITGA7*, *ITGBL1*, *ITGA6*, and *ITGA2*). These are generally most highly expressed in melanocytes, followed by endothelial cells, fibroblasts, smooth muscle cells (mainly *ITGA7*), keratinocytes and macrophages (Figure 4B). Integrin beta-2 (*ITGB2*) is highly expressed in macrophages, integrin alpha-11 (*ITGA11*) in fibroblast cells, and integrin beta-3 (*ITGB3*) and integrin alpha-10 (*ITGA10*) on endothelial cells. The expression profile in melanoma for the most expressed genes in normal skin tends to follow the same profile, except that integrin beta-4 (*ITGB4*) and integrin alpha-5 (*ITGA5*) have lower expression levels, and integrin alpha-3 (*ITGA3*) and integrin alpha-6 (*ITGA6*) have higher expression levels in melanoma (Figure 4).

Consistent with its role in metastatic melanoma, *ITGB3* is amongst the highest expressed genes in melanoma (Figure 3B) and, *ITGA10*, is highly expressed particularly in endothelial cells (Figure 4B). Indeed, the implication of integrins in angiogenesis in (metastatic) melanoma is nicely reflected by the finding that integrins generally are highly expressed in endothelial cells (Figure 4B).

## 4. Immunoglobulin-Like Cell Adhesion Molecules (IgCAMs)

IgCAM family is a large and diverse family with 59 members in humans (Appendix A). IgCAM protein structures are characterised by sandwiched anti-parallel β-sheets. Through these Ig-like domains, they facilitate both homophilic and heterophilic ligand binding in a calcium independent manner (Figure 5A). Unlike the cadherin and integrin families, IgCAMs do not provide much strength in their binding interactions, and they are predominantly involved in fine tuning adhesion, short term binding and tissue development [56]. The IgCAM family associated with nerve cells relay their intracellular signals from the microenvironment via the phosphorylation of tyrosines. Other IgCAMs, however, are transmembrane tyrosine phosphatases.

### 4.1. IgCAMs in Physiological Skin Context

IgCAM family member ICAMs are largely expressed on epithelial cells and utilise heterophilic binding. ICAM-1 (*ICAM1*) is physiologically present in low levels on leukocytes and endothelial cells, as well as melanoma cells, and levels can increase upon cytokine stimulation, e.g., with tumour necrosis factor (TNF) and interleukin (IL)-1. It can assist in leukocyte binding to epithelial cells and transmigration into the tissue [57]. CD146, another IgCAM member of significant interest in melanoma, under physiological conditions has a major role in development, especially in the vascularization of tissues such as in the kidney [58]. N-CAM (Neural cell adhesion molecule) (*NCAM1*) is one of the best studied IgCAM family members. Its physiological function is to provide cell adhesion. It is expressed on many cell types including nerve cells and usually facilitates cell–cell binding via homophilic interactions with the other cells. The family has an important role in development, as was seen when mice lacking N-CAM, developed with neural defects [59]. Like the other CAM families, the IgCAM family can also facilitate intracellular signalling, allowing for the ability to influence cell behaviour not only cell adhesion.

### 4.2. IgCAMs in Pathological Skin

Recently, ICAMs have been implicated in the enhanced cell migration and adhesion of melanoma cells onto endothelial cells via the STAT-3 pathway following stimulation from arginase-II, leading to increased STAT-3 activation and ICAM expression [10]. The increased cell adhesion and migration was supressed by the STAT-3 inhibitor static. The authors suggest that arginase II contributes to the migratory and adhesive capabilities of melanoma cells due to the upregulation of ICAM via the JAK/STAT pathway, which can lead to cell growth and survival of the melanoma cells [10]. Additionally, a recent study showed that that the increased ligation of ICAM-1 in the TME may increase the activation of STAT-3 through the toll-like receptor 4 (TLR4) via MYD88 and TRIF proteins [60]. The expression of TLR4 can be beneficial to the melanoma cells as they lead to immuno-suppression within the TME, resistance to apoptosis and cell survival [58]. In both studies the activation of STAT-3 was shown to lead to the increased tumour growth, angiogenesis and EMT as well as an immunosuppressive TME in mice in the latter study. This corroborates earlier findings showing that the expression of ICAM leads to the transepithelial migration of melanoma cells [61].

Neural cell adhesion molecule L1 (L1CAM; *L1CAM*) is a member of the IgCAM family and was previously shown to increase melanoma migration via the MAPK pathways [62] (see also Figure 5B). A later study, using a xenograft mouse model of human melanoma, showed that the knockdown of L1CAM reduced the melanoma cell’s ability to metastasise [63]. Gene expression arrays of the cell lines used in the study showed an increase in p53 and p38, which may have reduced the metastatic potential of the cells [63]. The intracellular domain of L1CAM was previously shown to bind to casein kinase 2 (CK2) and therefore inhibit the phosphorylation of PTEN and p53 and, thus, any ensuing damage response that may lead to apoptosis of the cancer cell [64]. L1CAM has high expression on melanoma cells, and the knockdown of L1CAM led to increased activity of p53/p21 and p38, significantly reducing the metastatic ability of the melanoma cells [63].

CD146 (cell surface glycoprotein MUC18; *MCAM*) is a well-known IgCAM member that plays an important role in melanoma progression and metastasis (see also Figure 6A). It was found to increase melanoma cell survival under stressful conditions through the AKT pathway, where CD146 and AKT were shown to have a reciprocal regulatory relationship by using pharmacological inhibitors [65]. CD146 has also been shown to influence the ezrin–radixin–moesin (ERM)-actin pathway, which further facilitates cell migration. CD146 physically binds to the ERM actin proteins and recruits further ERM proteins promoting extrusion and elongation. CD146 activated RhoA leads to increased RhoA activity and migration of the melanoma cells through the activation of the PIP5Ks/AKT pathway, a lipid kinase which plays a role in cancer cell evasion and cell survival through the PI3K/AKT pathway [66,67]. More recently, CD146 has been investigated as a potential prognostic marker and targeted for therapeutic intervention. A photo-immunotherapeutic approach was tested in a recent study showing a specific effect on melanoma cells, reducing cellular growth through production of ROS [68]. CD146 has also been shown to influence intracellular signalling, as with other CAMs, and the wide breath of its functions was previously reviewed by Xiyun Yan et al. [11].

The ability of the immunoglobulin-like cell adhesion molecules to affect intracellular signalling and increase migration and overall aggressiveness of the tumour, makes them attractive potential targets. Both, targeting to increase the efficacy of existing therapy by increasing the ability of tumour infiltrating leukocytes to access the tumour cells, and direct inhibition by novel therapeutics are possible strategies for treatment.

### 4.3. Gene Expression Profiles of IgCAMs in Normal Skin and Melanoma

The IgCAM family consists of 59 members with diverse functions in the skin and melanoma (Appendix A). The mRNA (pTPM) expression of this family in skin is spread across multiple cell types present in the skin (Appendix A; Figure 6B). L1CAM is highly expressed in melanocytes and has low expression in other cells. L1CAM and CD82 expression levels in melanocytes and melanoma cells follow a similar profile. The IgCAM family also have a role in the immune response as VSIG4 is almost exclusively expressed on macrophages and ICAM3 has a relatively high expression in T-cells (Figure 6B), and both have very low or no expression in melanoma cells (Appendix A). ESAM is a member, which is highly expressed in smooth muscle cells in physiological conditions and low levels in melanocytes, while in melanoma cells it is highly expressed.

## 5. Selectins in Cell Adhesion

Selectins are carbohydrate binding CAMs. There are three subclasses of selectins, endothelial (E)-selectin (SELE), leukocyte (L)-selectin (SELL) and platelet (P)-selectin (SELPLG). P and L selectin can bind sulphate glycans in a calcium independent manner (Figure 7A). The selectins usually play roles in homeostasis immune response and inflammation.

### 5.1. Selectins in Normal Skin

The selectin family plays an important role in physiological immune function within the epidermis allowing the infiltration of T-cells into the epidermis during development. The binding of selectins can also initiate the activation of integrin binding as well as an increase in chemokine expression [12]. During development, T cell precursors in the thymus express selectins that are present on the vascular endothelium [69], allowing the migration of the T-cell precursor out of the thymus. Following their arrival at the epidermis, they can bind to the keratocytes as they express high levels of the CCR10 receptor of the CCL27 ligand expressed by the keratinocytes [70]. Therefore, selectin expression and chemokine expression allow for the migration of the T-cell precursors to the epidermis. Selectins are one of the most under studied class of homing/cell adhesion molecules in cancer and thus, where they deviate from normal functions, can open novel possibilities for therapeutic targets.

### 5.2. Selectins in Pathological Skin

The selectin family plays a major role in the progression of melanoma, as well as initiating complications in cancer patients such as coagulation defects, leading to a poorer prognosis [12]. They mainly influence the melanoma cell progression through the microenvironment.

E-selectin has been shown to regulate the transepithelial migratory pathway through p38 and ERK kinases [12]. The loss of E-selectin can inhibit the T-cells and monocytes present in the epidermis from being activated against the melanoma cells. The expression can also allow the melanoma cells to progress and migrate to other parts of the body by binding to vascular endothelium and undergoing transepithelial migration via E-selectin, and through altering the tight junction modulator VE-cadherin (CDH5) [71]. In murine studies, the knockdown of E-selectin led to the decrease of liver metastasis, which had been induced experimentally [72]. This demonstrates the importance of this CAM molecule in the development and progression of melanoma, and cancer more generally, and is suggested to be an interesting target for therapeutic intervention (see also Figure 7B and Figure 8A and Appendix A).

As the melanoma cells promote angiogenesis of new blood vessels to facilitate the increased need for nutrients of the tumour [73], these blood vessels are often poorly constructed and often lack sufficient selectin expression on endothelial cells. This further allows for immune evasion as the leukocytes present in the blood cannot access the tumour as transepithelial migration without selectin expression does not occur. Different expression of selectin ligands on the tumour cells surface can induce different specific processes between cancer cells within a tumour due to the heterogeneity between tumour cells, and the amount of selectin expression is correlated with poorer patient outcomes [12].

L-selectin is known to be a mediator of the recruitment of leukocytes to the tumour microenvironment. Leucocytes can assist cancer cells in navigating through the endothelial barrier where angiogenesis may have stopped and thus facilitates migration and metastasis. These heterotypic interactions between cancer cells and the endothelial cells of the blood vessels can allow entry of the cancer cells into the blood stream [12]. More specific to melanoma, a recent study showed tumours containing L-selectin expressing T-cell respond better to immunotherapy. This increased benefit of L-selectin on T-cells is reportedly not due to T-cell homing, but rather the activation of the T-cells already presents within the melanoma [74].

P-selectin is another member of the CAM family that has an influence on the out-come of melanoma patients. P-selectin can mediate the interaction between cancer cells and platelets and neutrophils. This interaction can lead to increased inflammation through the increased expression of cytokines and chemokines. Platelet accumulation in the TME may hinder micro vascularization leading to a hypoxic microenvironment. Platelet deposition can also lead to increased thrombin deposition which itself can activate cancer cells and endothelium cells through the PAR signalling, which in turn can also lead to an increase in inflammation [12]. Despite this, it has been long established that P-selectin levels are low in primary and metastatic melanomas [75]. More recent studies however have implicated P-selectin in melanoma metastasis through the activation of acid sphingomyelinase in platelets [76]. Recent studies, such as this, show indirect and novel selectin involvement in melanoma, and further studies on its role in the surrounding TME could reveal novel ways to increase treatment efficacy or novel melanoma treatments, especially considering the selectin families role in migration of cells into the bloodstream.

### 5.3. Gene Expression Profiles of Selectins in Normal Skin and Melanoma

Of the three selectin members, E-selectin is most highly expressed in human skin, followed by P- and L-selectin (Appendix A; Figure 8B) [19,20]. E- and P-selectin are highest expressed in melanocytes and endothelial cells, while L-selectin is predominantly found in melanocytes and T-cells. Though, the selectin family is not often expressed on melanocytes/melanomas (Figure 8), the selectins expressed on the surrounding microenvironment cells (Figure 8B) are expected to have a significant influence on the progression and especially the migration/metastasis of the melanoma cells.

## 6. Conclusions

The current literature shows the importance of the cell adhesion molecules and their relationship with the microenvironment leading to the alteration of cell fate in melanocytes and melanoma cells. The loss and acquisition of different CAMs across the CAM families is an early event in melanoma, with the loss of anchorage on the basement membrane and between the surrounding keratinocytes. This allows the melanoma cells to proliferate and invade the dermis (Figure 9). Acquiring further mutations, the CAM profile of melanoma also changes, allowing for the degradation of the ECM and the ability to move through the ECM, and eventually allowing transepithelial migration into the bloodstream and metastasis.

All CAM families discussed in this review not only play a structural role in the adhesion of the cell to the surrounding cells or the ECM, but they also have the capacity to initiate cell signalling cascades. E-cadherin mediates the ability of melanocytes to proliferate by binding the β-catenin transcription factor, which in melanoma is released due to the loss of E-cadherin and enables the melanoma cells to proliferate. Notably, E-cadherin can also perform critical immune functions by attracting T-cells to induce cytotoxic antitumour functions. Thus, loss of E-cadherin contributes to melanoma progression on multiple levels. Understanding the signalling and transcriptional mechanisms that cause the expression loss of E-cadherin is critical in order to identify targets and treatment options that can alter these pathways to re-establish the expression of E-cadherin.

The integrin αVβ3 can affect the survival of melanoma cells through the new metastatic environments. The binding of this integrin can increase this survival through FAK and Pax1 activation downstream, which can increase the survival of the cells through AKT activation and the proliferation of the cells through β-catenin. The expression of this integrin is also associated with the disbalance to the antiapoptotic proteins BCL:BAX leading to increased cell survival. Indeed, integrins are attractive anti-melanoma drug targets, and several preclinical and phase I/II studies have been completed to date [77].

The activation of signalling pathways associated with IgCAMs increase the proliferation and migration of melanoma cells. ICAMs, through the activation of STAT-3, lead to increased tumour growth, angiogenesis, EMT, and an immunosuppressive TME. Multiple studies found that the increased L1CAM expression of melanoma cells leads to the inhibition of tumour suppressors such as p53 and PTEN, which are commonly downregulated in melanoma, thus allowing the escape from cell death in these cells. Thus, inhibition of L1CAM is another promising anti-melanoma target [61].

While these three families of CAMs lead to survival, metastasis, immune evasion, or growth of melanoma cells, the selectin family has an indirect effect on the ability of the melanoma to invade. The lack of selectins on poorly constructed tumour micro blood vessels allows an increase in immune suppression as the leukocytes in the bloodstream can-not attach to the endothelial cells of the blood vessels and infiltrate the tumour. Their expression can allow melanoma cells to enter distant organs via transepithelial migration.

The gene expression profiles of these CAMs taken from publicly available databases largely agree with the data from the literature. However, it also shows that CAMs are generally understudied with respect to their role in physiological skin and melanoma, as for the majority of CAM family members that are high expressed in skin, the functional roles have not been studied so far. The single cell RNAseq data discussed in this review provide an important quantitative view of the distribution of CAM among different cell types in the skin and their expression in melanoma versus normal melanocytes, which can provide insights and potentially novel avenues of research.

## Figures and Tables

**Figure 1 biomolecules-11-01213-f001:**
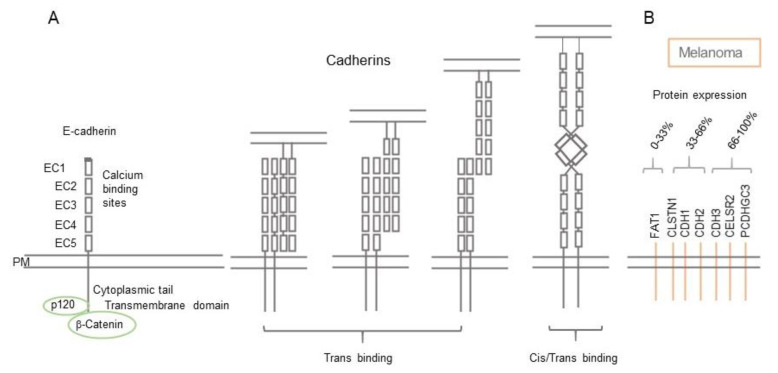
Schematic structure of cadherins and protein expression in melanoma. (**A**) Extracellular and intracellular structure of E-cadherin in *trans-* and *cis*-binding conformations. Trans binding refers to the cadherin molecules on one cell binding to cadherin molecules on other cells (cell-to-cell dimerization). Cis binding interfaces form between cadherins from the same cell surface and have been proposed to play a role in cadherin clustering. Intracellular signalling associated proteins are circled in green. (**B**) Cadherin protein expression on melanoma cells obtained from the Human Protein Atlas (immunohistochemistry staining). Staining intensities are estimated based on the percentage of patient samples positively stained for the respective CAM and classified into low (0–33% of patients), medium (33–66% of patients), and high expression (66–100% of patients). See also Appendix A.

**Figure 2 biomolecules-11-01213-f002:**
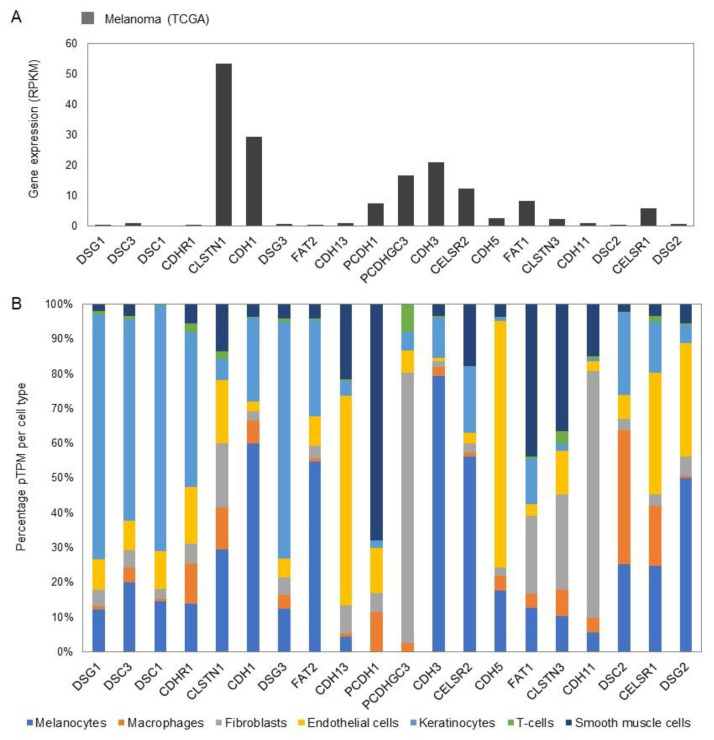
Gene expression of cadherins in melanoma cells and different cell types of healthy human skin. (**A**) Expression of cadherin genes in metastatic melanoma cells from single-cell RNA-seq experiments (Tirosh et al., 2016). (**B**) Expression of cadherin genes in different cell types of healthy human skin from single-cell RNA-seq experiments (Sole-Boldo et al., 2020), where expression values of clusters of similar cell types were merged. See also Appendix A.

**Figure 3 biomolecules-11-01213-f003:**
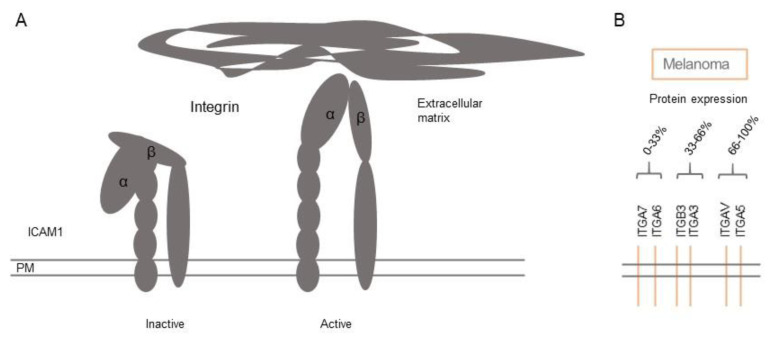
Schematic structure of integrins and protein expression in melanoma. (**A**) Extracellular structure of the ICAM 1 as an example for the integrin family. (**B**) Integrin protein expression on melanoma cells obtained from the Human Protein Atlas (immunohistochemistry staining). Staining intensities are estimated based on the percentage of patient samples positively stained for the respective CAM and classified into low (0–33% of patients), medium (33–66% of patients), and high expression (66–100% of patients). See also Appendix A.

**Figure 4 biomolecules-11-01213-f004:**
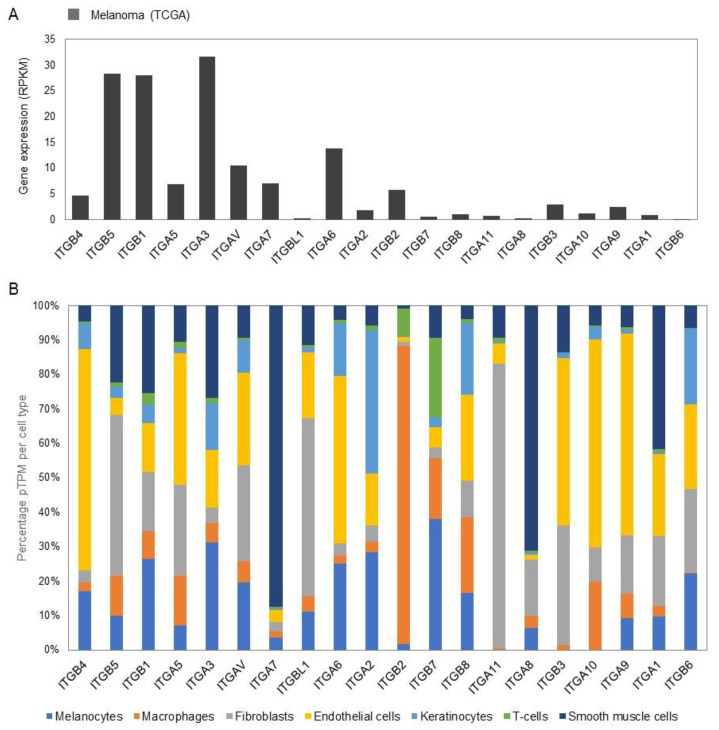
Gene expression of integrins in melanoma cells and different cell types of healthy human skin. (**A**) Expression of integrin genes in metastatic melanoma cells from single-cell RNA-seq experiments (Tirosh et al., 2016). (**B**) Expression of integrin genes in different cell types of healthy human skin from single-cell RNA-seq experiments (Sole-Boldo et al., 2020), where expression values of clusters of similar cell types were merged. See also Appendix A.

**Figure 5 biomolecules-11-01213-f005:**
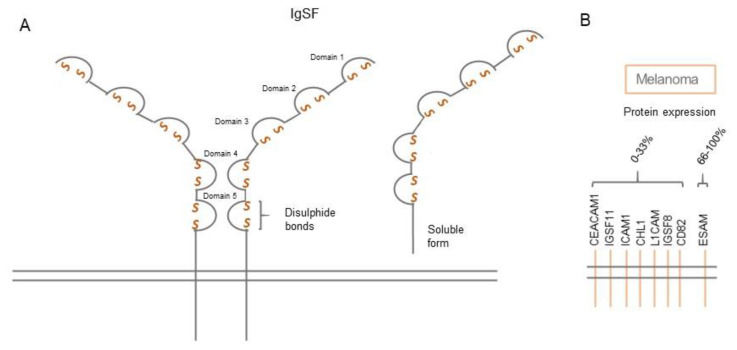
Schematic structure of immunoglobulins and protein expression in melanoma. (**A**) Extracellular structure of the immunoglobulin superfamily in the membrane bound and secreted form. (**B**) IgSF protein expression on melanoma cells obtained from the Human Protein Atlas (immunohistochemistry staining). Staining intensities are estimated based on the percentage of patient samples positively stained for the respective CAM and classified into low (0–33% of patients), medium (33–66% of patients), and high expression (66–100% of patients). See also Appendix A.

**Figure 6 biomolecules-11-01213-f006:**
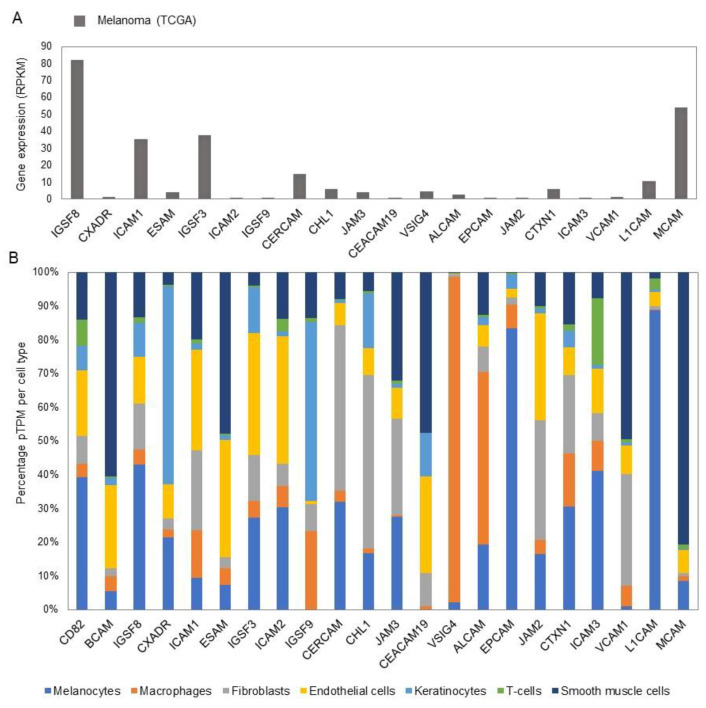
Gene expression of IgSF CAMs in melanoma cells and different cell types of healthy human skin. (**A**) Expression of IgSF CAM genes in metastatic melanoma cells from single-cell RNA-seq experiments (Tirosh et al., 2016). (**B**) Expression of IgSF CAM genes in different cell types of healthy human skin from single-cell RNA-seq experiments (Sole-Boldo et al., 2020), where expression values of clusters of similar cell types were merged. See also Appendix A.

**Figure 7 biomolecules-11-01213-f007:**
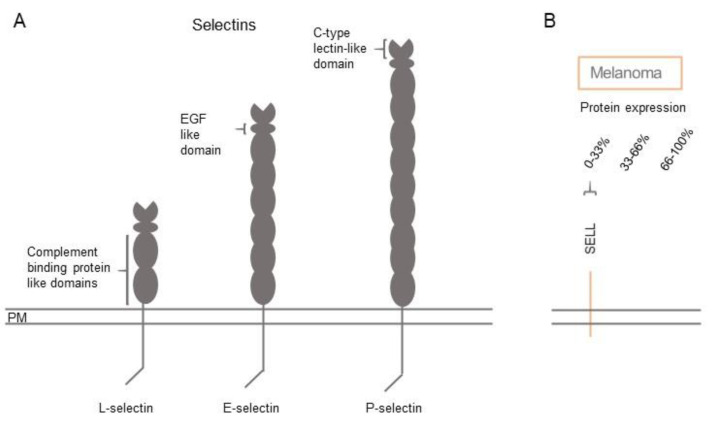
Schematic structure of selectins and protein expression in melanoma. (**A**) Extracellular representation of the extracellular binding domains of the selectin family. (**B**) Selectin protein expression on melanoma cells obtained from the Human Protein Atlas (immunohistochemistry staining). Staining intensities are estimated based on the percentage of patient samples positively stained for the respective CAM and classified into low (0–33% of patients), medium (33–66% of patients), and high expression (66–100% of patients). See also Appendix A.

**Figure 8 biomolecules-11-01213-f008:**
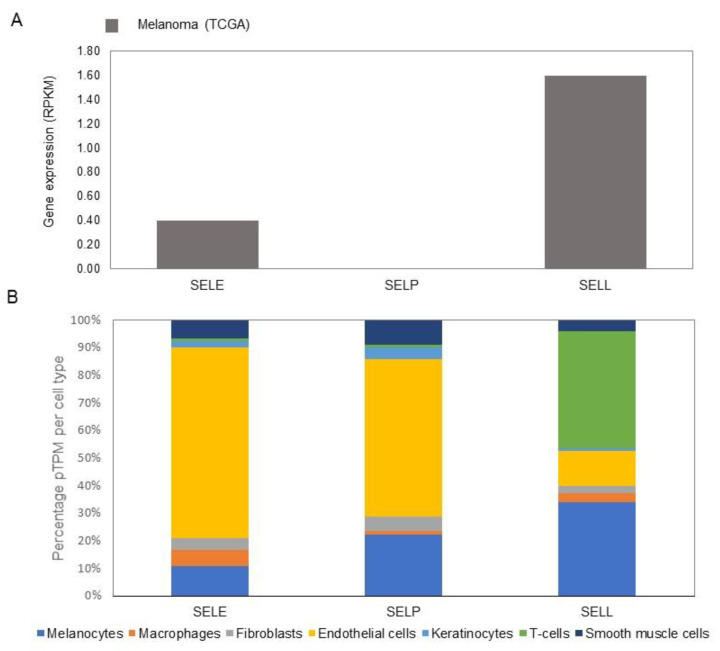
Gene expression of selectins in melanoma cells and different cell types of healthy human skin. (**A**) Expression of selectin genes in metastatic melanoma cells from single-cell RNA-seq experiments (Tirosh et al., 2016). (**B**) Expression of selectin genes in different cell types of healthy human skin from single-cell RNA-seq experiments (Sole-Boldo et al., 2020), where expression values of clusters of similar cell types were merged. See also Appendix A.

**Figure 9 biomolecules-11-01213-f009:**
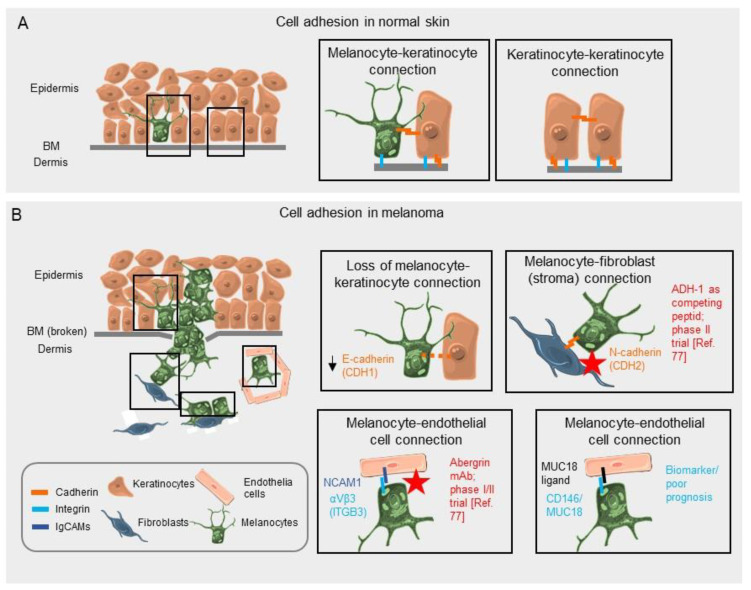
Mechanism of CAMs in normal skin and melanoma. (**A**) Schematic representation of CAMs in physiological skin. The melanocyte-BM connection is facilitated by integrin α6β1; keratinocytes connect via E-cadherin; keratinocytes and melanocytes connect via E-cadherin; and the keratinocytes-BM connection works through integrin α5β1, α3β1, α2β1, and α6β1 (different integrins bind different anchors i.e., collagen, laminins, ECM on the BM). (**B**) Schematic representation of CAMs in melanoma and indication of markers and drug targets. The loss of E-cadherin impacts the keratinocyte-melanocyte connection; expression of N-cadherin favours the connection of melanoma cells with stromal cells; expression of ITGB3 on melanoma cells favours connection to endothelial cells; and MUC18 expression enables the interaction with endothelial cells. The red star indicates that the interaction is a target for anti-melanoma drugs. The figure was prepared using images from Servier Medical Art by Servier (http://www.servier.com/Powerpoint-image-bank (accessed on 9 August 2021)), which is licensed under a Creative Commons Attribution 3.0 Unported License.

**Table 1 biomolecules-11-01213-t001:** Examples of members of the four CAM families and their physiological and pathological roles.

Family	Protein Name (Gene ID)	Physiological (Melanocyte) Role	Pathological (Melanoma) Role	Reference
Cadherin	E-cadherin (CDH1)	Maintains homeostasis within the epidermal melanin unit via the cell to cell adhesion of the keratinocytes and melanocytes	Loss of expression allows for the increased proliferation and increased mobility	[4,5]
	N-Cadherin (CDH2)	Cell–cell adhesion in fibroblasts in the dermis	Facilitates preferential binding to fibroblasts in the dermis and poorer prognosis, potential therapeutic target	[6,7]
Integrin	Integrin beta 7 (ITGB7)	Highly expressed in melanocytes	Not expressed in melanoma	Figure 4a,b
	αVβ3 (ITGB3)	Low expression in melanocyte	Expressed in melanoma cells increasing proliferation, survival and metastasis	[8,9]
Immunoglobulin superfamily	ICAM 1 (ICAM1)	Expressed on endothelial and immune cells	Expressed on melanoma cells increasing cell migration and growth	[10]
	CD146 (MCAM)	Not expressed in melanocytes	Expressed in metastatic melanoma and used a prognostic marker	[11]
Selectin	E-Selectin (SELE)	Low expression in melanocytes	Facilitate immune evasion via lack of expression on poorly constructed tumour vesicles	Figure 8a,b, [12]
	L-selectin (SELL)	Not expressed in melanocytes	Low expression in melanoma can aid in migration through epithelial barrier	[12]

## Data Availability

All data used in this manuscript are publicly available and are included in the Appendix A.

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
