# Peer review of "Cell Adhesion Molecules in Normal Skin and Melanoma"

_biomolecules, 2021, doi:10.3390/biom11081213_

Round 1

Reviewer 1 Report

Dear Authors,

thank you for your good review.

I only missed one thing:

Please provide an overview of the different CAMs familes as a table!

Familiy  -  important members  -  short summary physiological role in particular skin ( - short summary pathophysiological role particullary melanoma (up/down regulation) - citation of most important paper/studies

Thank you!

Author Response

Response to Reviewer 1 Comments

Dear Authors,

thank you for your good review.

I only missed one thing:

Point 1: Please provide an overview of the different CAMs familes as a table!

Familiy  -  important members  -  short summary physiological role in particular skin ( - short summary pathophysiological role particullary melanoma (up/down regulation) - citation of most important paper/studies

Thank you!

Response 1: We appreciate this suggestion and have provided such a table now.

Reviewer 2 Report

The manuscript is well written and brings the reader up to date. Only a few ideas for improvement.

  1. The sentence lane 60: CAMs are important in the context of melanoma as they can influence the ability of the immune system to gain proximity to the tumour or to recognise the tumour

is very interesting. Unfortunately, these immunology aspects were not   picked up in detail, later in the  text.

  1. Regarding all CAMs therapeutic aspects could be discussed; that is a further interesting aspect for melanoma researchers.

  1. Chaper 2.3 and Figure 2 are great highlights of the manuscript. It would be nice to include some primary literature about rare/unusual cadherins in melanoma.

Author Response

Response to Reviewer 2 Comments

The manuscript is well written and brings the reader up to date. Only a few ideas for improvement.

Point 1: 1. The sentence lane 60: CAMs are important in the context of melanoma as they can influence the ability of the immune system to gain proximity to the tumour or to recognise the tumour is very interesting. Unfortunately, these immunology aspects were not   picked up in detail, later in the  text.

Response 1: While we already picked up these immunology aspects (e.g. in lines 196-208 and 523-534), we agree that this is an important point; we highlight this now again in the conclusion section.

Point 2: 1. Regarding all CAMs therapeutic aspects could be discussed; that is a further interesting aspect for melanoma researchers.

Response 2: We appreciate this comment. While we already mention therapeutic aspects in the text (e.g. for Cadherins in lines ~246-274, Integrins lines ~421-434, IgsF ~541-545 and for selectins only mentioned briefly line 620), we agree that this is important and we pick this up again in the conclusion section where we cite a recent review that summarizes therapeutic approaches based on CAMs. We also include examples in a new figure 9.

Point 3: 2. Chaper 2.3 and Figure 2 are great highlights of the manuscript. It would be nice to include some primary literature about rare/unusual cadherins in melanoma.

Response 3: It was actually surprising to see how understudied many of the CAMs are in skin and melanoma context, despite their high expression levels. We mention this now in the conclusion section.

Reviewer 3 Report

Overall, the manuscript entitled “Cell Adhesion Molecules in Normal Skin and Melanoma” is a very well documented review on molecular mechanisms of action which are involved this set of molecules and propose several potential biomarkers for detection and treatment of metastatic melanoma. Specific comments: - Abstract: Good concision and clarity. - Introduction: this chapter has enough information about the subject of the study. - The cadherin family mediating cell-cell attachment: complete explanation about the role of cadherins. In paragraph "2.3. Gene expression profiles of cadherins in normal skin and melanoma" there were several hyphens in words which don't need it: Tran-scripts (246), ex-pressed (250), cadher-in-13 (251), ex-pressed (253), desmo-collin-1 (257). Please, correct them. - Integrins in cell-ECM adhesion: This chapter is well written and clearly exposed, except in paragraph "3.1. Integrins in physiological skin context" where new not needed hyphens were found: al-lowing (303), be-haviour (316). Please, correct them. - Immunoglobulin like cell adhesion molecules (Ig CAMs): as previous chapter, arguments are well and clearly exposed, and only a few not needed hyphens were found in paragraphs "4.2. Ig CAMs in pathological skin context" and "4.3. Gene expression profiles of IgCAMs in normal skin and melanoma": in-crease (466), ex-pression (491) - Selectins in cell adhesion: concise chapter with enough explanations. - Conclusions: in addition to arguments exposed in this chapter, that seems another abstract instead of a proper conclusions of the review, this reviewer encourage authors to mention their own opinions about most relevant items extracted from the text and what adhesion molecules are more promissing to use as biomarkers. - Tables: Supplementary tables are very appropiate. - Figures: Figures are very illustrative. Only some little details need a correction. In figure 1, a green remarked circle (as mentioned in caption) is missing. In figure 5B, protein expression at level range 33-66% are not clearly marked, giving place to a misinterpretation. Finally, in my opinion, I missing a figure showing mechanism of action where are involved CAMs in nomal skin and melanoma. I highly recommend to include such figure in manuscript.

Author Response

Response to Reviewer 3 Comments

Overall, the manuscript entitled “Cell Adhesion Molecules in Normal Skin and Melanoma” is a very well documented review on molecular mechanisms of action which are involved this set of molecules and propose several potential biomarkers for detection and treatment of metastatic melanoma. Specific comments:

- Abstract: Good concision and clarity.

- Introduction: this chapter has enough information about the subject of the study.

Point 1: - The cadherin family mediating cell-cell attachment: complete explanation about the role of cadherins. In paragraph "2.3. Gene expression profiles of cadherins in normal skin and melanoma" there were several hyphens in words which don't need it: Tran-scripts (246), ex-pressed (250), cadher-in-13 (251), ex-pressed (253), desmo-collin-1 (257). Please, correct them.

Response 1: We have corrected those mistakes.

Point 2: - Integrins in cell-ECM adhesion: This chapter is well written and clearly exposed, except in paragraph "3.1. Integrins in physiological skin context" where new not needed hyphens were found: al-lowing (303), be-haviour (316). Please, correct them. 

Response 2: We have corrected those mistakes.

Point 3: Immunoglobulin like cell adhesion molecules (Ig CAMs): as previous chapter, arguments are well and clearly exposed, and only a few not needed hyphens were found in paragraphs "4.2. Ig CAMs in pathological skin context" and "4.3. Gene expression profiles of IgCAMs in normal skin and melanoma": in-crease (466), ex-pression (491) - Selectins in cell adhesion: concise chapter with enough explanations. 

Response 3: We have corrected those mistakes.

Point 4: - Conclusions: in addition to arguments exposed in this chapter, that seems another abstract instead of a proper conclusions of the review, this reviewer encourage authors to mention their own opinions about most relevant items extracted from the text and what adhesion molecules are more promissing to use as biomarkers. 

Response 4: We have rewritten the conclusion section and also include a new figure 9 that includes CAMs that are most promising as biomarkers and anti-melanoma drug targets.

- Tables: Supplementary tables are very appropiate.

Point 5: Figures: Figures are very illustrative. Only some little details need a correction. In figure 1, a green remarked circle (as mentioned in caption) is missing. In figure 5B, protein expression at level range 33-66% are not clearly marked, giving place to a misinterpretation. 

Response 5: We have corrected those figures.

Point 6: Finally, in my opinion, I missing a figure showing mechanism of action where are involved CAMs in nomal skin and melanoma. I highly recommend to include such figure in manuscript.

Response 6: We appreciate this suggestion. We have now prepared such as figure; we also use this figures to highlight therapeutic opportunities.